

# Soil microbial diversity in organic and non-organic pasture systems

Mohan Acharya[1], Amanda J. Ashworth[2], Yichao Yang[3], Joan M. Burke[4], Jung Ae Lee[5] and Roshani Sharma Acharya[6]

[1] Department of Animal Science, University of Arkansas at Fayetteville, Fayetteville, AR, United States of America
[2] Poultry Production and Product Safety Research Unit, United States Department of Agriculture, Agricultural Research Service, Fayetteville, AR, United States of America
[3] Department of Crop, Soil, and Environmental Science, University of Arkansas at Fayetteville, Fayetteville, AR, United States of America
[4] United States Department of Agriculture, Agriculture Research Service, Dale Bumpers Small Farms Research Center, Booneville, AR, United States of America
[5] Agriculture Statistics Lab, University of Arkansas at Fayetteville, Fayetteville, AR, United States of America
[6] Entomology and Plant Pathology, University of Arkansas at Fayetteville, Fayetteville, AR, United States of America

Corresponding author
Mohan Acharya, macharya@uark.edu

## ABSTRACT

Understanding the effects of organic pasture management on the soil microbiome is important for sustainable forage production since soil microbiome diversity contributes to improved nutrient cycling, soil structure, plant growth, and environmental resiliency; however, the soil microbiome response to pasture management is largely unknown. This study assessed the soil microbial diversity, richness, and community structure following 10 years of pasture management (organic or non-organic) of the V4 region of the 16S rRNA using the Illumina MiSeq platform. Soil samples were collected from 0–15 cm in July and August from 2017–2018 and soil nutrient properties (nutrients, carbon, nitrogen, and pH) quantified and correlated with soil microbial diversity. Overall, greater soil bacterial species richness ($P \leq 0.05$) occurred in organic relative to non-organic (conventional) systems. Management affected bacterial species richness (Chao1), with greater richness occurring in organic pasture soils and less richness occurring in non-organic systems ($P \leq 0.05$). Similarly, management affected bacterial evenness (Simpson's index), with a more diverse community occurring in organically managed soils relative to non-organic pastures ($P \leq 0.05$). Linear discriminant analysis effect size analysis showed statistically significant and biologically consistent differences in bacterial taxa in organic compared with non-organic soils. Therefore, there was a shift in bacterial community structure in organic relative to non-organic soils ($P \leq 0.05$). Additionally, soil nutrients (Fe, Mg, Ni, S, Al, K, Cd, and Cu), pH, C, and N were correlated with one or more dominant bacterial phyla (Gemmatimonadetes, Planctomycetes, Firmicutes, Chloroflexi, Actinobacteria, and Acidobacteria). Overall, pasture management affected soil microbial diversity, with greater diversity occurring in organic than non-organic systems, likely owing to applications of organic poultry litter in organic systems compared to non-organic management (use of inorganic-fertilizers and herbicides). Results indicate that when pastures are converted to organic production systems, soil microbial richness and diversity may increase, thereby resulting in enhanced soil microbiome diversity and overall ecosystem services.

## INTRODUCTION

Soils contain highly diverse microorganisms, with $10^9$ to $10^{10}$ microorganisms per gram (*Gans, Wolinsky & Dunbar, 2005*). Soil bacterial species and diversity are important as they are major decomposers of soil organic matter and can influence carbon and nitrogen turnover rates (*Hättenschwiler, Tiunov & Scheu, 2005*), which provides substrates to microbes and nutrients to plants. Nitrogen mineralization in the soil also depends on the abundance and diversity of bacterial taxa (*De Ruiter et al., 1993*). Higher bacterial diversity is crucial for the breakdown of organic matter that involves complex processes such as chitin degradation (*Beier & Bertilsson, 2013*). Human activity in soils, including increased use of chemical fertilizers, pesticides, and tillage, has led to a decrease in global soil biodiversity (*Hirsch, 2010*). Other negative anthropogenic environmental impacts include soil degradation, pesticide accumulation, reduced water availability, and increased greenhouse gas emission, collectively hampering biodiversity (*Foley et al., 2005*). Conventional farming systems, crop monocultures, use of chemical fertilizers, and intensive use of agrochemicals eliminate certain group of microbes and decrease overall soil microbial diversity (*Stagnari et al., 2014*).

Organic production practices reduce the use of chemical fertilizers and pesticides and increase sustainable production practices (*Gomiero, Pimentel & Paoletti, 2011*). Organic soil amendments such as manure, may increase soil microbial diversity, richness, and community structure (*Chaudhry et al., 2012*; *Lupatini et al., 2017*). However, there are still incomplete understandings of the effects of organic production practices on soil biodiversity (*Wu & Sardo, 2010*; *Tscharntke et al., 2012*; *Bünemann, Schwenke & Van Zwieten, 2006*). Although, DNA sequencing technologies have made it possible to identify soil bacterial abundance and diversity (*Soliman et al., 2017*).

Over one-quarter of the Earth's total land area is used for livestock grazing (*Herrero et al., 2013*). Grazing directly or indirectly impacts soil microbial communities and C and N availability because livestock feeds on aboveground plants and returns half of the grazed forage to the soil as feces, changes plant composition, and shifts rhizosphere exudation (*Lovell, Jarvis & Bardgett, 1995*; *Manzano & Návar, 2000*; *Le Roux et al., 2008*; *Yang et al., 2013*; *Tardy et al., 2015*). Animal manures in pasture increase soil enzyme activity, nutrient availability, provides food sources for soil microbes, and increases soil bacterial diversity (*Ashworth et al., 2017*; *Das et al., 2017*). A recent study by *Yang et al. (2019)* found that animal inputs (poultry litter and cattle manure) increased soil microbial diversity and richness by altering the soil nutrient status. Earlier studies also demonstrated grazing pressure impacts soil microbial communities, with positive associations occurring for soil productivity and microbial α-diversity, and the suppression of bacterial communities in overgrazed soils (*Xun et al., 2018*).

Pasture soils are less managed than row crops and thus have greater potential microbial stability (*Lauber et al., 2009*; *Ashworth et al., 2017*). Limited studies are available on the impact of organic pasture management on soil microbial communities. Therefore, the objective is to determine how 10 years of livestock pasture management (organic or conventional) impacts soil microbial diversity.

## MATERIALS AND METHODS

### Study management history

The research was conducted at the USDA-ARS Dale Bumpers Small Farms Research Center in Booneville, Arkansas (35.08°N, 93.98°W), where plots with different management, i.e., organic and non-organic, were established and consistently managed since 2007. Thirtytwo-hectare plots utilized for sheep were managed organically since 2007 and certified organic in 2012 (Nature's International Certification Services, Viroqua, WI). Forage type was predominantly tall fescue (*Festuca arundinacea*). Non-organic sheep pastures were utilized by sheep for more than 30 years and were predominantly bermudagrass (*Cynodon dactylon*). Both forage types have limitations for sheep production and provide little pollinator habitat. Before plots were established, sheep grazed at approximately ∼10 sheep ha$^{-1}$ according to forage availability, which was seasonal and dependent on forage species present (*Eremopyrum triticeum, Lolium perenne* and other voluntary forbs and grasses). Details regarding the soil types of the study area are available in earlier publication from the research site (*Thomas et al., 2008*). The site received 101 cm rainfall in 2017 and 129 cm in 2018, with the 30-year (1987–2017) mean annual precipitation of 126 cm (Fig. 1A). Mean annual temperature in 2017 and 2018 were 19.1 and 14.8 °C, with a winter minimum of 9.5 °C and summer maximum of 32.3 °C (Fig. 1B). Additionally, daily rainfall and temperature (*NOAA, 2017/2018*), for 3 days before sample collection is provided in the Table 1.

### Pasture plot preparation for non-organic forage plots

Two 0.16-ha non-organic plots were used for the non-organic treatment in this study. Non-organic plots were sprayed with Roundup [N-(phosphonomethyl)glycine)] (41% glyphosate, 4.67 l ha$^{-1}$) in June, July, September, and October of 2016 and January of 2017, and with Outrider [N-[[(4,6-dimethoxy-2-pyrimidinyl)amino]carbonyl]-2-(ethylsulfonyl)imidazo[1,2-a]pyridine-3-sulfonamide;{Sulfosulfuron}] (Monsanto, St. Louis, MO; 0.096 l ha$^{-1}$) in September 2016 using a Continental Belton cluster nozzle sprayer (Continental Belton McAlester, SR: A44117, Oklahoma city, OK). Non-organic plots were tilled (Maschio Gaspardo North America Inc., SC 300, Des Moines, IA), and rolled using 12' Big Guy Roller (Grahl Manufacturing, St. Louis, MO) in October 2016. Long-term (15 years) applications of inorganic nitrogen (ammonium nitrate) were applied to pastures at recommended rates (https://www.uaex.edu/publications/PDF/FSA-2153.pdf). The seed-mixes were Buck's Hangout (Hamilton Native outpost, Elk creek, MO; http://www.hamiltonnativeoutpost.com/14.5 kg ha$^{-1}$) used in half of each 0.16 ha plot, and other Tallgrass Inexpensive or Tallgrass Exposed Clay subsoil mix (Prairie Moon, Winona, MN; www.prairiemoon.com; 13.44 kg ha$^{-1}$ and 26.8 kg ha$^{-1}$, respectively according to

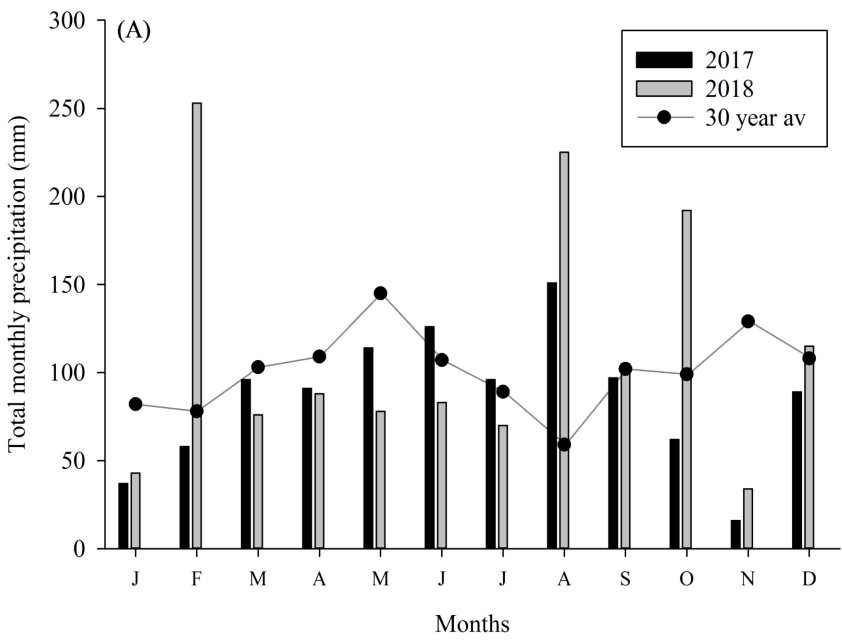

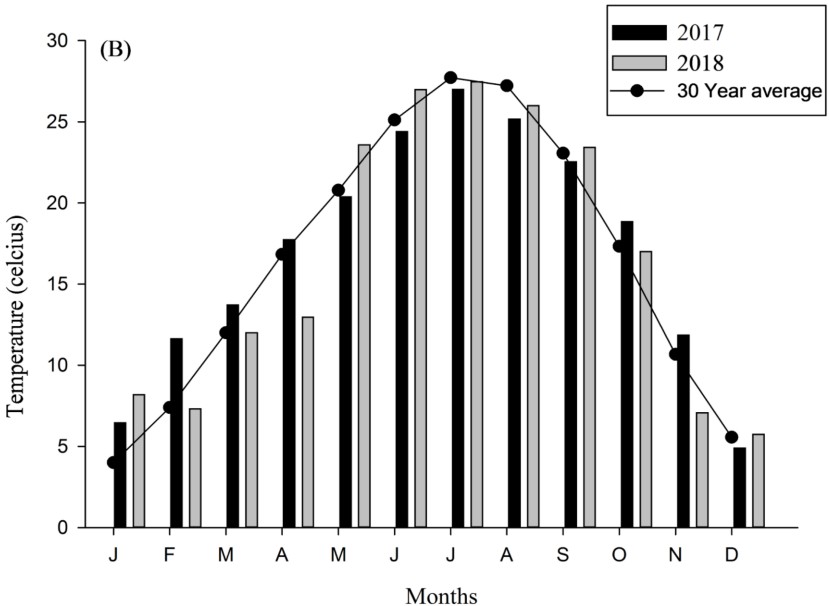

**Figure 1** **Temperature and rainfall data from study site.** (A) Total monthly precipitation and (B) mean monthly temperature in Booneville, AR in 2017 and 2018. Total monthly precipitation and temperature (1987–2017; 30 Year average) were obtained from NOAA (2018).

**Table 1** Daily rainfall and temperature for 3 days before the sample collection in 2017 and 2018 in Booneville, AR.

| | 2017 | | | | | | 2018 | | | | | |
| | July | | | August | | | July | | | August | | |
| | 2 d[a] | 1 d | 0 d | 2 d | 1 d | 0 d | 2 d | 1 d | 0 d | 2 d | 1 d | 0 d |
|---|---|---|---|---|---|---|---|---|---|---|---|---|
| Temperature (°C) | 29.7 | 34.3 | 35.5 | 23.7 | 31.2 | 34.4 | 35.9 | 32.1 | 37.0 | 32.6 | 34.2 | 34.9 |
| Rainfall (mm) | 14.2 | 0.0 | 0.0 | 5.2 | 0.0 | 0.0 | 0.0 | 3.6 | 0.0 | 0.0 | 0.0 | 0.0 |

Notes.
[a]d, number of days before sample collection.

commercial nursery recommendations) in the other half of each 0.16 ha plot, planted on February 7, 2017, and again same time in 2018 (species percentage per seed mixes are available in the aforementioned websites).

## Organic pasture plot preparation

Plots were controlled burned using forestry drip torch after establishing a firebreak around the plots. A cover crop of oats (*Avena sativa*) was planted in the winter before spring planting native seed mixes. One, 0.4 ha organic plot was seeded on February 3, 2017, with "Butterfly and Hummingbird" Mix from Hamilton Native outpost (Elk Creek, MO; www.hamiltonnativeoutpost.com/; at 8.5 kg ha$^{-1}$; species composition and percentage of each species are available in the above website). The other organic plot was 0.8 ha planted with Tall Grass Inexpensive (Prairie Moon Nursery; 13.4 kg ha$^{-1}$). The seed was drilled onto a prepared seed bed using a Brillion planter (Brillion Farm Equipment, Sure Stand Model SSP-8, Brillion, WI) at a 1.25 cm depth. Soil fertility was adjusted using poultry litter (4.4 Mg ha$^{-1}$; fresh weight basis) following the University of Arkansas soil test recommendation (https://www.uaex.edu/publications/PDF/FSA-2153.pdf).

## Sheep grazing

Plots were multifunctional for pollinator habitat and sheep forage. Sheep grazed plots if they became weedy (including *Chenopodium album* and *Ambrosia artemisiifolia*), or stubble height reached 1 m. Grazing of the conventional plots occurred to remove excess forage and weeds, but not to the detriment of the native plant species. The stocking rate of each plot was different based on visual assessment of forage availability, which included plant height, the density of unwanted, and desired plants.

## Soil sample collection

Soil samples were collected in July and August of 2017 and repeated again in 2018. Soil samples were collected from 0–15 cm, and soil probes were sterilized between plots with 70% ethanol. Samples were packed in sterile whirl-pack bags and stored at −20 °C until analysis.

Soil sample collection sites were georeferenced using a Global Positioning System unit. For the analysis, 32 independent samples were taken (2 per pasture system × 2 sub-samples per treatment × 2 replications × 2 sample dates per year × 2 years).

## Soil physiochemical analyses

Soil samples were dried at 70 °C for 48 h and then ground with mortar and pestle to pass through a two mm sieve. Mehlich-3 extractable nutrients (i.e., Al, As, B, Ca, Cd, Co, Cr, Cu, Fe, K, Mg, Mn, Mo, Na, Ni, P, Pb, S, Se, Ti, and Zn) were determined using 1:10 soil volume: extractant solution volume ratio (*Tucker, 1992*) and analyzed by inductively coupled argon-plasma spectrometry using a 7300 IPC-OES DV (Perlin-Elmer, Waltham, MA). Total C and N were analyzed via dry combustion using a Vario Max CN combustion analyzer (Elementar Americas Inc., Mt. Laurel, NJ). Soil pH was determined on 1:1 soil: distilled water suspension using a symphony B30PCI probe (VWR International, Atlanta, GA).

## DNA extraction, PCR amplification, and sequencing

Most of the methods adopted here are previously described in *Gurmessa et al. (2021)*. Briefly, DNA was extracted from each soil samples using MpBio Fast DNA Spin kit (MpBio Laboratories, SKU 116560200-CF) following manufacturer's protocol. DNA thus extracted was quantified using Quant-ItTM PicoGreen® (Invitrogen) dsDNA quantitation assay and stored at −20 °C until further analysis. Bacterial community composition and diversity was determined through classification of 16S rRNA gene by Illumina MiSeq sequencing platform. The V4 region of prokaryotic 16S rRNA genes were amplified with barcoded primers 515F and 806R (*Caporaso et al., 2011*). Libraries for each samples were pooled and sequenced in parallel, and 291 base-paired end sequence were obtained resulting in total 3,531,936 sequence reads. Raw sequences were processed in Mothur software v.1.40.0, using a Miseq SOP protocol (*Kozich et al., 2013*). Sequences that did not match the primers were eliminated from demultiplexed sequence reads. Sequence reads were clustered into OTUs (Operational Taxonomic Units) at 97% similarity threshold. After the quality control pipeline, 2,906,534 sequence reads remained while remaining were deleted.

## Data analysis and statistics

The green genes database was used to classify the OUT at the genus level using the Bayesian method (*Cole et al., 2009*), after that relative abundance of all OTUs was summed within phylum and analyzed for the relative abundance of OTUs at the phylum level. Subsampled data was used to calculate Chao, Shannon and Inverse Simpson index via Mothur and compared results using ANOVA in the statistical software R 3.5.1 (*R Core Team, 2013*) and JMP R 12 (*SAS Institute, 2015*). Weighted and unweighted UniFrac distance, and Bray-Curtis were used to define bacterial beta-diversity. Principal coordinate analysis based on weighted and unweighted UniFrac with permutational analysis of variance (PERMANOVA) as statistical methods were carried out to compare bacterial communities at the phylum OUT level (*Clarke & Gorley, 2006*; *Dhariwal et al., 2017*). Linear Discriminant Analysis Effect Size measurement was used to identify taxa differences between treatments with Galaxy (*Segata et al., 2011*).

Spearman correlation was used to find correlation among soil pH, C, N, C: N ratio, soil mineral (Al, As, B, Ca, Cd, Co, Cr, Cu, Fe, K, Mg, Mn, Mo, Na, Ni, P, Pb, S, Se, Ti, Zn) and number of OTU of most dominant bacteria phyla per treatment (organic and

non-organic system). Management (organic or non-organic) and time of sample collection (July or August) was used as the fixed effect and year and replications as the random effect. Principal component analysis was used to determine relationship among variables using R software. Proportion of variance was done to find variance explained by each PCs. Three PCs that explained most variance were selected for further analysis. The eigenvectors were used to show directional orientation (positive or negative) of all variates with respect to the centroid both in numbers and in figures using PCA biplots.

## RESULTS

### Bacterial community composition based on treatment, year, and sampling period

There were differences in soil bacterial communities at the phyla level between sampling years (2017 and 2018; $P \leq 0.05$); however, there were no differences in bacterial communities between non-organic and organic treatments and sampling period [July and August ($P \geq 0.05$); Table 2]. The following top ten phyla dominated soil bacterial communities: Proteobacteria (mean relative abundance of all libraries was 33.13% in 2017 vs. 32.63% in 2018), Bacteriodetes (6.43% in 2017 vs. 33.44% in 2018), Acidobacteria (23.00% in 2017 vs. 6.86% in 2018), Actinobacteria (8.75% in 2017 vs. 7.07% in 2018), Verrucomicrobia (6.78% in 2017 vs. 8.51% in 2018), Chloroflexi (5.75% in 2017 vs. 3.64% in 2018), Firmicutes (5.95% in 2017 vs. 3.29% in 2018), Planctomycetes (5.90% in 2017 vs. 1.74% in 2018), Gemmatimonadetes (3.26% in 2017 vs. 0.55% in 2018), and Saccharibacteria (1.06% in 2017 vs. 2.28% in 2018; Fig. 2A). The top 20 soil bacterial communities at the OTU level are presented in Fig. 2B.

### Bacterial community alpha diversity is influenced by treatment, sampling years, and sampling period

Alpha diversity was influenced by treatment (organic vs. non-organic), year, and sampling period, as estimated by three different algorithms (Chao1, Shannon, and Simpson's index). Specifically, management affected bacterial species richness (Chao1), with greater richness found in organic pasture soils and fewer species richness occurring in non-organic soils ($P \leq 0.05$) (Table 3). Species richness also varied by years, with greater richness occurring in 2017 compared to 2018 ($P \leq 0.05$). Richness differences between July and August were at the margin of statistical significance ($P = 0.056$). There were richness differences for treatment × year ($P \leq 0.05$). The Simpson's index was used to calculate the evenness of soil bacterial community and indicated treatments affected bacterial evenness, with a more diverse community occurring in organically managed soils, relative to non-organic soils ($P \leq 0.05$). Year also influenced bacterial community evenness, with greater evenness found in 2018 compared to 2017 ($P \leq 0.05$). However, the sampling period did not result in evenness differences ($P \geq 0.05$). The treatment × year interaction affected bacterial community evenness ($P \leq 0.05$). The Shannon index was used to measure the community diversity, including both richness and evenness, and there were no treatment or sampling period differences ($P \geq 0.05$). The treatment × year interaction affected bacterial community diversity ($P \leq 0.05$). Figure 3A illustrates the influence of treatment and sampling period

**Table 2 PERMANOVA of bacterial community structure by treatment (organic and non-organic soil), year (2017 and 2018), and sampling period (July and August).** PERMANOVA results illustrate differences in bacterial community structure by single factors of management (organic and non-organic), sample collection years (2017 and 2018), and sampling period (July and August), as well as two factors (treatment × year, treatment × sampling period, and year × sampling period), and three factors (treatment × year × sampling period).

| Factor | Pseudo-F | P-value |
|---|---|---|
| Treatment | 0.84 | 0.498 |
| Year | 15.26 | 0.001 |
| Sampling period | 0.84 | 0.491 |
| Treatment × year | 9.55 | 0.001 |
| Treatment × sampling period | 1.62 | 0.293 |
| Year × sampling period | 1.99 | 0.096 |
| Treatment × sampling period × year | 0.82 | 0.465 |

on soil bacterial richness using Chao1, with Fig. 3B showing bacterial evenness (calculated by using Simpson index), and Fig. 3C presenting the bacterial community diversity (via Shannon index).

## Bacterial community structure following treatments and sampling periods

Community structures across treatments and sampling periods were compared to determine if treatments (non-organic and organic) and sampling periods (July and August) impacted the bacterial community structure. Bray-Curtis distance matrix was used to calculate pairwise distances from all samples. Visualization of these distances using a PCoA revealed an overlap between bacterial communities from treatments and sampling periods. Significant shifts in community structure from non-organic and organic soils were observed in PCoA plots based on Bray-curtis (PERMANOVA R = 0.183, $P \leq 0.05$; Fig. 4). Non-organic samples were distinct from organic samples; however, samples collected from July and August were similar, especially in non-organic systems.

The linear discriminant analysis effect size was used to identify bacterial features represented between non-organic and organic soils and between July and August. The log LDA score cutoff was set at 2 (as an absolute value). Both the negative and positive values for the cutoff can be considered to determine significant features when the score is lower than 2. LEfSe analysis confirmed two features (OTU38 and OTU44), showing statistically significant and biologically consistent differences in non-organic soil samples collected from August. The other two significant features (OTU11 and OTU56) were found in organic soil samples collected in July (Fig. 5). These identified bacterial species reflected distinct abundance in soil samples. LEfSe also detected 11 bacterial species showing statistically significant differences in organic soil samples collected from August. However, no significant feature can be found in samples collected from non-organic soil in July (Fig. 5).

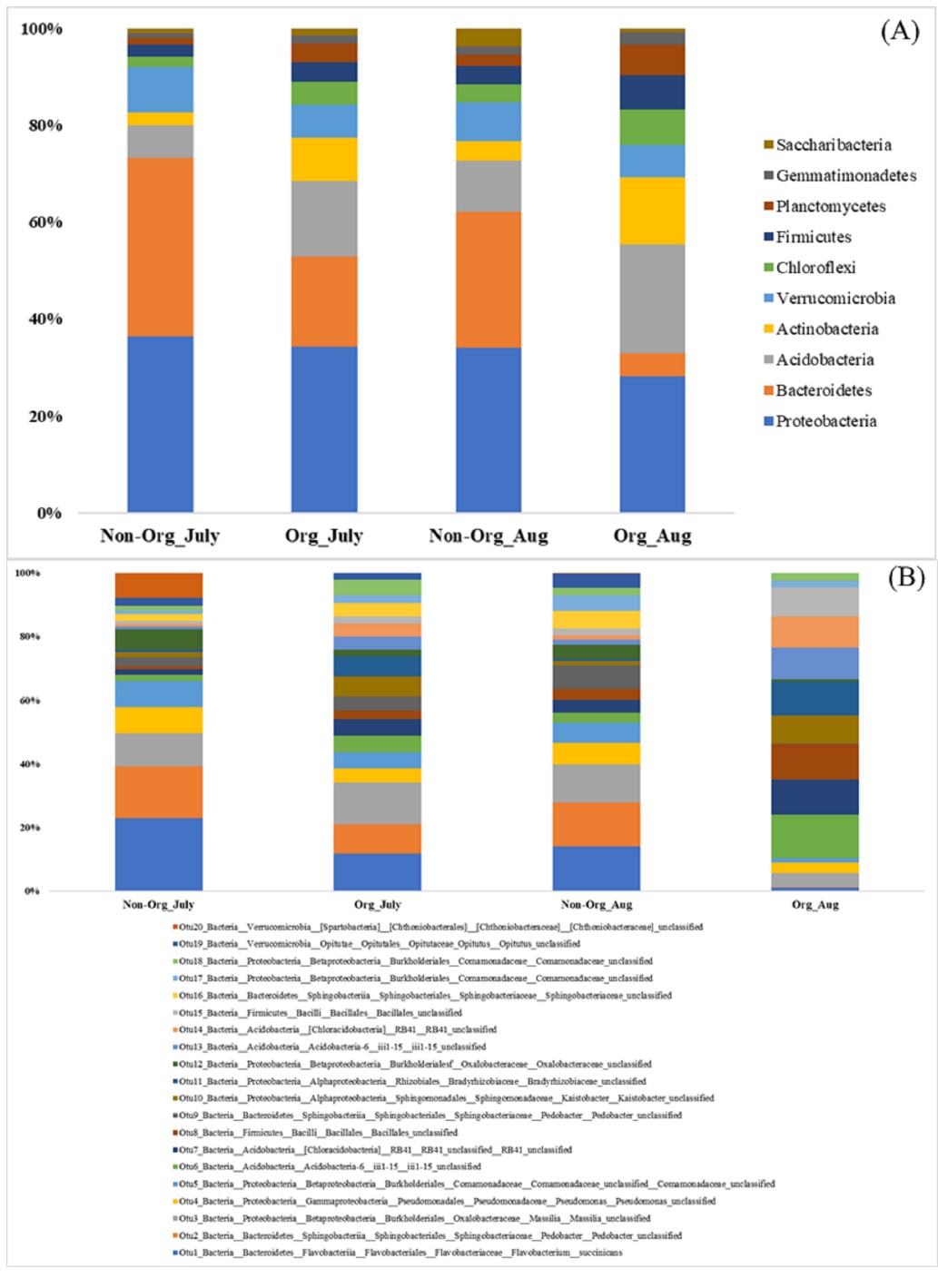

**Figure 2** **Proportion of soil bacteria in phylum and OTU level by different pasture management system and sampling period.** Mean relative proportion of soil bacteria in phylum level (A) and in OTU level (B) by treatment and sampling period from 2017-2018. Treatments include non-organic (Non-Org) soil and organic (Org) soil. The sampling period includes July and August (Aug). The order of colors is the same in the legend as the bars.

**Table 3** **ANOVA of richness, evenness, and diversity in bacterial community structure of treatment, year and sampling period.** ANOVA results illustrating richness, evenness and diversity in bacterial community structure by single factor of treatment (non-organic and organic), year (2017 and 2018) and sampling period (July and August), as well as two factors (treatment and year).

| Parameter | Factor | F-value | P-value |
|---|---|---|---|
| Richness | Treatment | 6.58 | 0.0156 |
| | Year | 10.37 | 0.0031 |
| | Sampling period | 3.95 | 0.0561 |
| | Treatment × Year | 4.52 | 0.0425 |
| Evenness | Treatment | 4.08 | 0.0525 |
| | Year | 11.39 | 0.0021 |
| | Sampling period | 2.01 | 0.1664 |
| | Treatment × Year | 8.14 | 0.0081 |
| Diversity | Treatment | 3.78 | 0.0613 |
| | Year | 13.77 | 0.0008 |
| | Sampling period | 2.01 | 0.1666 |
| | Treatment × Year | 13.33 | 0.0011 |

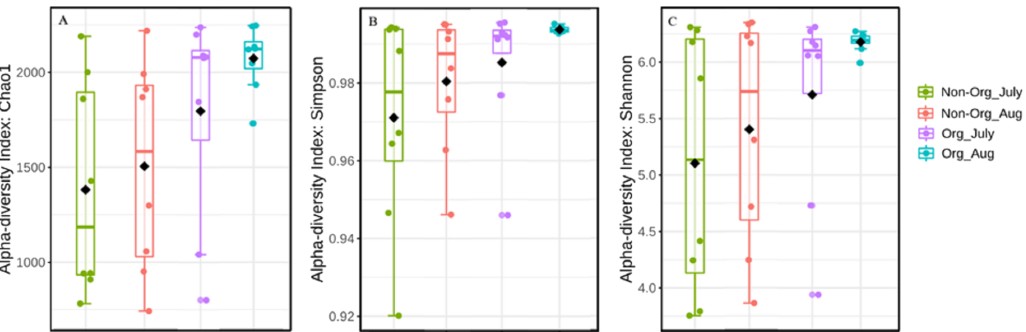

**Figure 3** **Bacterial richness, evenness, and diversity in different pasture management system.** Mean soil bacterial richness, evenness, and diversity [Chao1 (A), Simpson index (B), and Shannon index (C)] in non-organic and organic pasture management system. Treatments include non-organic (Non-org) and organic (Org) soils. Soil samples were collected during two sampling periods (July and August).

## Soil chemical property relationship with bacterial community structure

The first three PCs were selected (Table 4) for further analysis based on their cumulative variability accounting for 65.80% of the total variation. Principal component 1 explained 37.42% of overall variance, while PC2 and PC3 contributed an additional 16.12% and 12.26%, respectively. Principal component 1, was most strongly affected by soil N, C, and S contents, which are correlated in Fig. 6, and in the biplot (Figs. 7A, 7B). All these factors are positive eigenvectors in PC1 which is shown numerically in Table 5, as well as in Figs. 7A and 7B (as all these vectors were directed toward a positive direction with respect to the centroid). Principal component 2 was most strongly associated with soil pH, Ca, and P contents. These factors are negative eigenvectors in PC2, which is shown in Table 5, as

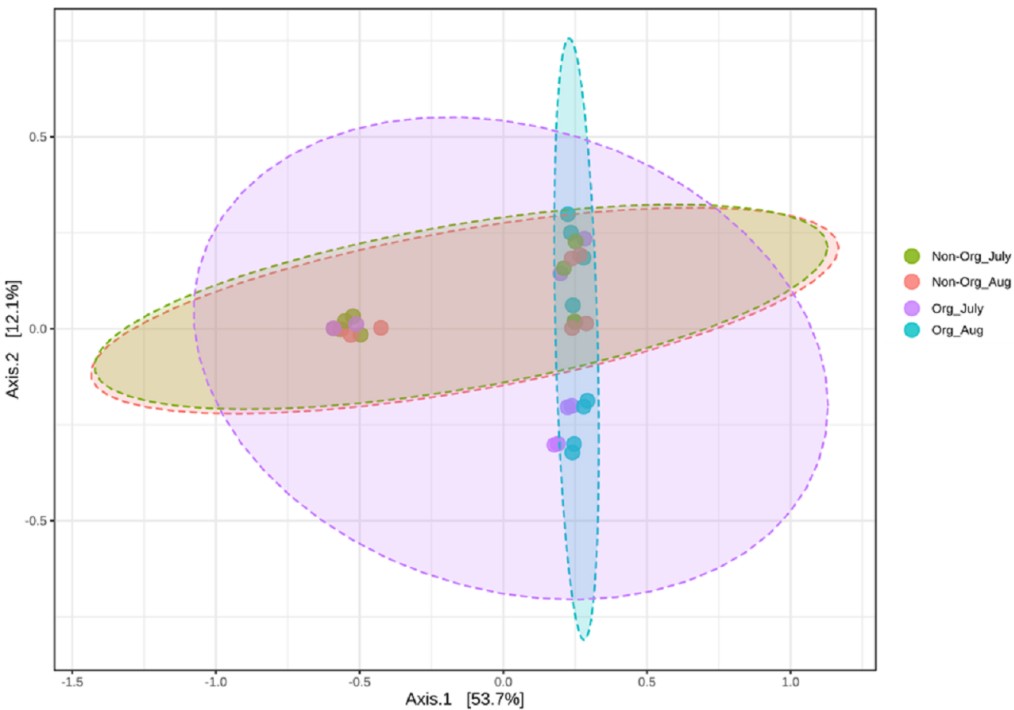

**Figure 4  Principal Coordinated Analysis (PCoA) of Bray–Curtis distances of bacterial community structures in different pasture management system and sampling period.** Principal Coordinated Analysis (PCoA) of Bray–Curtis distances of bacterial community structures in different pasture management. The factors include non-organic (Non-Org) soil and organic (Org) soil collected during two different sampling periods (July and August).

well as in Figs. 7A and 7C as these vectors were directed toward the negative side. Likewise, PC3 was closely related to soil Zn content, Gemmatimonadetes, and Acidobacteria, which are seen in Table 5 and Figs. 7B and 7C.

Soil pH was negatively correlated with Actinobacteria ($R^2 = -0.61$), Acidobacteria ($R^2 = -0.50$), Chloroflexi ($R^2 = -0.45$), and Planctomycetes ($R^2 = -0.48$). Nitrogen and C were positively correlated with Gemmatimonadetes (N, $R^2 = 0.63$; C, $R^2 = 0.67$), Planctomycetes (N, $R^2 = 0.78$; C, $R^2 = 0.81$), Firmicutes (N, $R^2 = 0.69$; C, $R^2 = 0.71$), Chloroflexi (N, $R^2 = 0.45$; C, $R^2 = 0.45$), Actinobacteria (N, $R^2 = 0.60$; C, $R^2 = 0.57$) and Acidobacteria (N, $R^2 = 0.68$; C, $R^2 = 0.72$). The significant correlation between OTUs of dominant bacterial phyla and soil minerals is given in Table 6. Likewise, relationship between soil pH, C, N, C:N ratio, minerals (Al, As, B, Ca, Cd, Co, Cr, Cu, Fe, K, Mg, Mn, Mo, Na, Ni, P, Pb, S, Se, Ti, Zn) and number of OTU of most dominant bacteria phyla can be seen in Figs. 5 and 7A–7C.

# DISCUSSION

## Soil microbial diversity in long-term organic vs non-organic pastures

The study was conducted to assess the microbial diversity, richness and community structure using 16S rRNA sequencing between soil samples taken from organic and

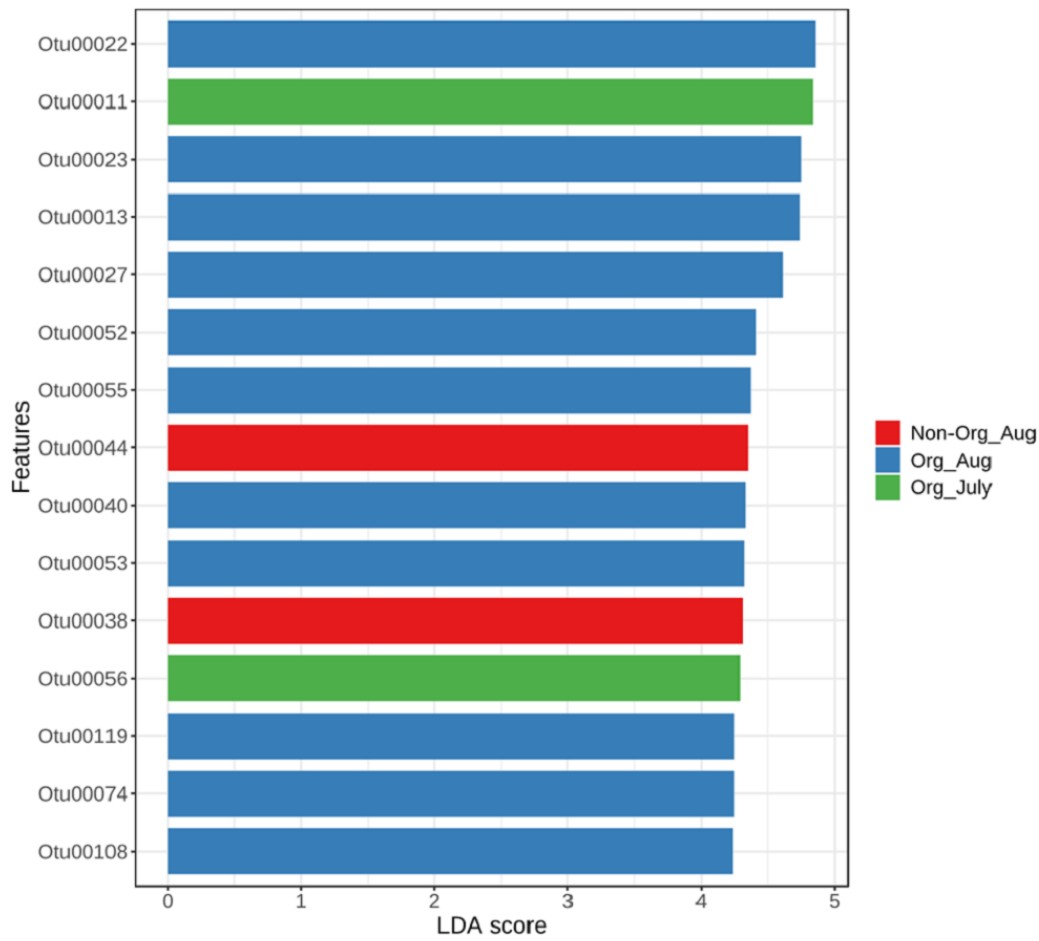

**Figure 5** **Linear Discriminant Analysis Effect Size (LefSe) analysis showing OTU abundance for different pasture management system and time period.** Linear Discriminant Analysis Effect Size (LefSe) analysis showing OTU abundance for different groups in non-organic soils collected from August and organic soils collected from July and August. No significant feature was observed in samples collected from non-organic soils in July.

**Table 4** **Standard deviation, proportion of variance and cumulative proportion of each principal component.** Standard deviation is the deviation of principal components. Proportion of variance explained how much variance is explained by each of the principal components with respect of the whole (the sum).

|  | PC1 | PC2 | PC3 |
| --- | --- | --- | --- |
| Standard deviation | 3.566 | 2.341 | 2.041 |
| Proportion of variance | 0.374 | 0.161 | 0.122 |
| Cumulative proportion | 0.374 | 0.535 | 0.658 |

non-organic pasture systems. Long-term organic or non-organic pasture management affected soil microbial diversity, richness, and community structure. Soil bacterial species richness (Chao1) was greater in organic in comparison to non-organic pastures. Likewise, greater bacterial evenness (Simson's index) occurred in organic relative to non-organic

## Spearman Correlation coefficients

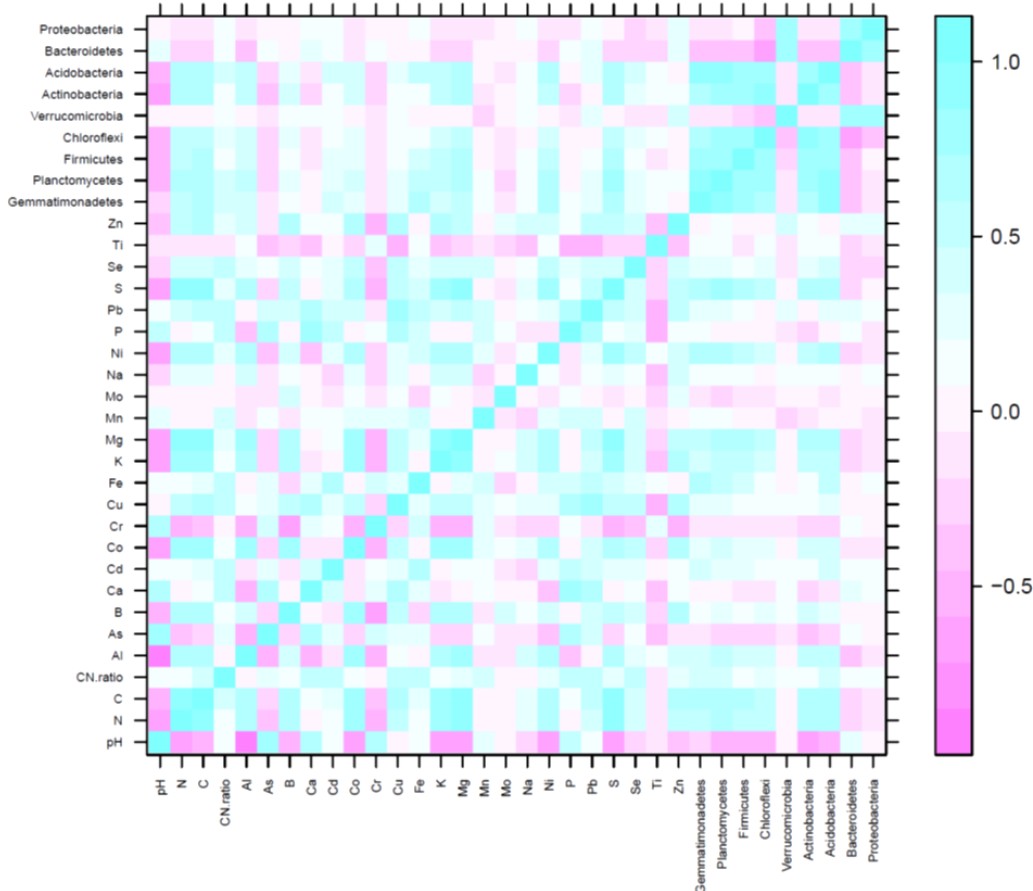

**Figure 6** **Spearman correlation coefficient of soil variables (pH, N, C and minerals) and number of operational taxonomic unit (OTU) of major bacteria phyla.** Blue color represents positive correlation, white color shows no correlation and pink color shows negative correlation.

pasture, and finally, greater microbial diversity was found in organic than non-organic pastures. Recent publications also suggests that organic management supports greater microbiome assemblages than non-organic systems (*Chen et al., 2020*). The greater microbial diversity in organically managed soil in the current study is consistent with other studies (*Hartmann et al., 2015*; *Mäder et al., 2002*). Higher diversity of microbial communities in organically managed pastures is likely related to the restricted use of chemical fertilizers and pesticides and with applications of poultry litter, which provides essential macronutrients for microbial growth and function (*Sun, Deng & Raun, 2004*; *Chaudhry et al., 2012*; *Ashworth et al., 2017*; *Yang et al., 2019*).

Increases in microbial diversity cause a more heterogeneous distribution of bacterial species, leading to healthier plant-soil associations and ecosystems benefits for improved sustainability (*Brussaard, De Ruiter & Brown, 2007*; *Crowder et al., 2010*). In non-organic pastures, herbicides and chemical fertilizers are regularly applied, causing long-term stress

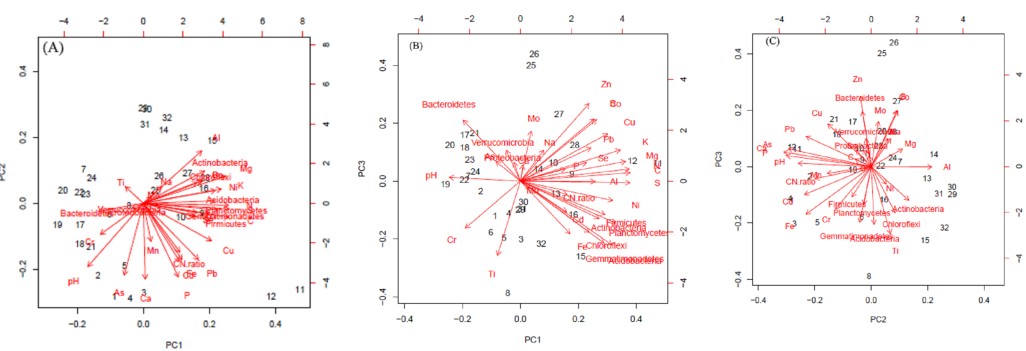

**Figure 7 Biplot of soil components (pH, C, N C: N and minerals) and bacteria phyla.** In (A) the horizontal axis shows the projection on to the principal component 1 (PC1) and the vertical axis of PC2. In (B) and (C), vertical axis shows projection on to PC3 while horizontal axis shows projection on to the PC3 and PC2 respectively.

to microbial populations leading to less diverse microorganisms, in part, because these chemicals potentially inhibit or eliminate certain groups of bacteria and select few bacterial phyla that could thrive in non-organic management systems (*El Fantroussi et al., 1999*; *Stagnari et al., 2014*).

In the current study, organic pastures were grazed by certified organic sheep (Nature's International Certification Services) and non-organic pastures by non-organic sheep. Pastures were regularly soil tested and continuously applied with animal manure (organic pastures) or synthetic fertilizers (in non-organic pastures), thus soil nutrients were never limited. A high nutrient availability could have dominated the presence of copiotrophic bacteria such as Proteobacteria, Firmicutes, Gemmatimonadetes, and Bacteroidetes (*Ryckeboer et al., 2003*; *Fierer, Bradford & Jackson, 2007*; *Fierer et al., 2012*; *Zhang et al., 2017*), accounting for more than 60% of the total bacteria phyla. Similarly, other oligotrophic microorganisms, such as actinobacteria and acidobacteria (*Fierer, Bradford & Jackson, 2007*) were also presented but in a relatively lower percentage. All of the bacterial communities were dominated by a few major phyla including: Proteobacteria, Bacteriodetes, Acidobacteria, Actinobacteria, Verrucomicrobia, Chloroflexi, Firmicutes, Planctomycetes, Gemmatimonadetes, and Saccharibacteria. The relative abundance of the most dominant bacterial taxa in this study is similar to that reported elsewhere (*Davinic et al., 2012*; *Lupatini et al., 2017*). Although, soil samples were taken to compare differences in bacterial taxa between organic and non-organic pastures with different management systems, it is important to recognize that most dominant bacteria taxa were largely the same in both treatments and months, with some differences only in their relative abundance.

It is generally thought that grazing affects bacterial species mainly through fecal and urine deposition (*Ritz et al., 2004*), by altering plant composition (*Wang et al., 2001*), and shifting rhizosphere exudation (*Guitian & Bardgett, 2000*). Several studies have reported the influence of plant species on soil bacterial communities (*Smalla et al., 2001*; *Schmidt et al., 2019*). Also, soil microbial communities are shaped not only by plant species, but

**Table 5  Eigenvectors of principal component (PC) representing a linear combination of the original variable for soil bacterial phylum and soil properties collected 0–15 cm in 2017 and 2018.**

|  | PC1 | PC2 | PC3 |
|---|---|---|---|
| pH | −0.173 | **−0.300** | 0.015 |
| N | **0.265** | −0.009 | 0.051 |
| C | **0.267** | −0.066 | 0.035 |
| C:N ratio | 0.113 | −0.231 | −0.057 |
| Al | 0.181 | 0.249 | −0.001 |
| As | −0.059 | −0.034 | 0.083 |
| B | 0.179 | 0.105 | 0.264 |
| Ca | 0.005 | **−0.360** | 0.068 |
| Cd | 0.112 | −0.274 | −0.132 |
| Co | 0.185 | 0.108 | 0.262 |
| Cr | −0.133 | −0.147 | −0.200 |
| Cu | 0.212 | −0.178 | 0.201 |
| Fe | 0.120 | −0.267 | −0.224 |
| K | 0.243 | 0.068 | 0.132 |
| Mg | 0.256 | 0.129 | 0.083 |
| Mn | 0.024 | −0.180 | −0.031 |
| Mo | 0.026 | 0.029 | 0.212 |
| Na | 0.057 | 0.080 | 0.131 |
| Ni | 0.224 | 0.061 | −0.082 |
| P | 0.109 | **−0.349** | 0.051 |
| Pb | 0.171 | −0.268 | 0.142 |
| S | **0.266** | −0.026 | −0.006 |
| Se | 0.161 | −0.062 | 0.078 |
| Ti | −0.055 | 0.079 | −0.317 |
| Zn | 0.167 | −0.044 | **0.330** |
| Gemmatimonadetes | 0.202 | −0.047 | **−0.264** |
| Planctomycetes | 0.232 | −0.030 | −0.178 |
| Firmicutes | 0.205 | −0.079 | −0.142 |
| Chloroflexi | 0.167 | 0.099 | −0.218 |
| Verrucomicrobia | −0.034 | −0.022 | 0.132 |
| Actinobacteria | 0.191 | 0.154 | −0.159 |
| Acidobacteria | 0.222 | 0.011 | **−0.271** |
| Bacteroidetes | −0.139 | −0.036 | 0.259 |
| Proteobacteria | −0.015 | −0.033 | 0.079 |

also by soil amendment (*Schmidt et al., 2019*), therefore, this study focused on organic or non-organic soil amendment on livestock pastures.

## Relationship between soil pH, C, N, soil C: N ratio and soil minerals on most dominant bacteria phyla

Although bacteria varied temporally and spatially, overall bacterial diversity was largely predicted by soil pH (*Lauber et al., 2009*). The effect of pH on bacterial community was evident at the phyla level where there was a high relative abundance of Gemmatimonadetes,

**Table 6  Pearson correlation between soil bacteria (Gemmatimonadetes, Planctomycetes, Firmicutes, Chloroflexi, Actinobacteria, and Acidobacteria) and soil nutrient concentration (Fe, Mg, Ni, S, Al, K, Cd, Cu) collected 2017 and 2018 to a 0–15 cm depth.**

| Bacterial phyla | Soil minerals | | | | | |
|---|---|---|---|---|---|---|
| Gemmatimonadetes | Fe, $R^2 = 0.66$ | Mg, $R^2 = 0.51$ | Ni, $R^2 = 0.58$ | S, $R^2 = 0.72$ | | |
| Planctomycetes | Al, $R^2 = 0.51$ | Fe, $R^2 = 0.60$ | K, $R^2 = 0.52$ | Mg, $R^2 = 0.65$ | Ni, $R^2 = 0.65$ | S, $R^2 = 0.81$ |
| Firmicutes | Cd, $R^2 = 0.51$ | Cu, $R^2 = 0.56$ | K, $R^2 = 0.53$ | Mg, $R^2 = 0.56$ | S, $R^2 = 0.73$ | |
| Chloroflexi | K, $R^2 = 0.53$ | Mg, $R^2 = 0.54$ | | | | |
| Actinobacteria | Al, $R^2 = 0.56$ | K, $R^2 = 0.58$ | Mg, $R^2 = 0.67$ | S, $R^2 = 0.58$ | | |
| Acidobacteria | Al, $R^2 = 0.52$ | Fe, $R^2 = 0.56$ | K, $R^2 = 0.52$ | Mg, $R^2 = 0.63$ | Ni, $R^2 = 0.66$ | S, $R^2 = 0.75$ |

Planctomycetes, Actinobacteria, Acidobacteria, and Chloroflexi, changing in a consistent manner across the soil pH gradient as depicted both by Spearman correlation as well as PCA biplots. Though some bacterial taxa are correlated with soil pH, pH may not be considered the universal predictor of microbial diversity. The effect of pH on bacterial community structure and diversity in the current study has been supported by various studies where different sequencing methods were used (*Blagodatskaya & Anderson, 1998*; *Bååth & Anderson, 2003*; *Cookson et al., 2007*). Specifically, relative abundance of Acidobateria, Actinobacteria, Planctomycetes, and Chloroflexi were correlated with soil pH, similar to that in the current study (*Sait, Davis & Janssen, 2006*; *Eichorst, Breznak & Schmidt, 2007*). It is notable that soil pH and bacterial community structure were correlated; however, it is hard to identify specific mechanisms that cause these relationships. There are two general explanations that may explain why soil pH was the predictor of bacterial community composition. First, most bacterial taxa have intracellular pH close to neutral (*Madigan & Martinko, 2005*) and are unable to survive if soil pH falls outside the narrow range of tolerance, i.e., pH of 6.5–7.5). Lower soil pH or acidification inhibits bacterial enzyme activity and overall cell mechanics (*Beales, 2004*), but some microorganisms can survive in acidic environments (*Beales, 2004*). Second, pH acts as an integrating factor for several soil variables, e.g., soil organic C, N, P, ion sorption, and salinity, and poultry litter applications are known to increase pH due to its liming effect (*Brady & Weil, 1999*).

Soil microbial communities affect C and N turnover with complex interactions involving vegetation types and various soil properties (*Bailey, Smith & Bolton Jr, 2002*; *Deng et al., 2016*). Therefore, soil microbial community abundance and diversity have significant correlations with soil total N and C (*Liu et al., 2012*). In the current study, a strong positive correlation with bacteria phyla (Gemmatimonadetes, Planctomycetes, Firmicutes, Chloroflexi, Actinobacteria, and Acidobacteria) and N and C was observed (both Spearman correlation and PCA biplots). This is important to understand the relationship of soil microbes and for sequestration and stability of N and C in grassland soils.

Close contact of soil minerals and soil microorganisms is essential for biogeochemical cycling and transformation of plant nutrients, and by acquiring necessary enzyme cofactors through minerals (*Jansson, 1987*; *Bennett et al., 2001*; *Semrau, DiSpirito & Yoon, 2010*; *Ahmed & Holmström, 2014*; *Jones & Bennett, 2014*). Yet, very limited studies have found a relationship between soil microorganisms and minerals. In the current study, some relationships between dominant soil microbiomes and minerals were observed.

Correlations existed between some soil minerals and dominant bacterial taxa. However, other factors such as pH could have masked the effect, and thus, a future controlled study may be needed to confirm this relationship.

**Effect of moisture on soil bacteria**

Precipitation drives bacterial abundance (*Clarholm & Rosswall, 1980*); therefore, rainfall data for three days before the sample collection are presented (Table 1). In the current study, Firmicutes and Actinobacteria (both gram-positive bacteria) were able to maintain their relative percentages even in a comparatively drier 2018 compared to 2017. It is because gram-positive cell wall structure allows these microorganisms to withstand dry conditions (*Schimel, Balser & Wallenstein, 2007*; *Lennon et al., 2012*; *Moreno-Espíndola et al., 2018*). Other factors, such as the presence of soil organic C and bacteria's ability to produce biofilm, also help increase or decrease the abundance of gram-negative bacteria (*Malik et al., 2018*; *Moreno-Espíndola et al., 2018*).

Soil microorganisms depend on organic C for their growth. Plants provide C to rhizosphere microorganisms in the form of exudates, secretions, lysates, and dead cells (*Singh et al., 2004*). However, during drought, the quantity and quality of C available to soil microbes negatively impact gram-negative bacteria as these are more tightly connected to freshly assimilated plant C (*Denef et al., 2009*). Drastic decreases of Acidobacteria, Planctomycetes, and Gemmanimonadetes in 2018 could be due to their gram-negative bacterial structure, making them vulnerable to dry soil (*Denef et al., 2009*). However, some gram-negative bacteria increased, which could be due to biofilms. Though gram-negative by structure, Proteobacteria, Chloroflexi, and Saccharibacteria were able to maintain similar percentages during 2018 compared with 2017, likely due to their ability to produce biofilms, thus helping them to withstand drought (*Or, Phutane & Dechesne, 2007*).

## CONCLUSIONS

The current study was performed to assess soil microbial diversity, richness, and community structure in organic and non-organic livestock pastures using 16S rRNA sequencing. Most of the dominant bacterial taxa were similar in both management systems with some differences in their relative abundance; however, greater soil microbial diversity was observed in organic livestock pastures. More heterogeneous distribution of bacterial species leads to healthier associations at the soil-plant-animal nexus and ultimately improves sustainable agriculture production. Though pH was correlated with the most dominant bacterial phyla, soil properties such as C and N were more strongly correlated. Additionally, we observed a correlation between several soil minerals (Fe, Mg, Ni, S, Al, K, Cd, and Cu) and bacterial phyla (Gemmatimonadetes, Planctomycetes, Firmicutes, Chloroflexi, Actinobacteria, and Acidobacteria). Limited studies are available that outline the relationship between soil minerals and bacterial phyla. However, soil minerals are essential for enzymatic reactions, biochemical cycling, production of chelating agents, and catalyzing reactions. Overall, greater soil microbial diversity in the organic system, suggests organically produced poultry litter, less herbicide and antibiotic use (during poultry and sheep production) resulted in greater soil microbial diversity than the conventional sheep

system. Further studies are needed to ascertain driving soil factors for microbial abundance to optimize soil function in the largest land-use category in the US, or grasslands.

**Abbreviations**

| | |
|---|---|
| **C** | Carbon |
| **N** | Nitrogen |
| **Al** | Aluminum |
| **As** | Arsenic |
| **B** | Boron |
| **Ca** | Calcium |
| **Cd** | Cadmium |
| **Co** | Cobalt |
| **Cr** | Chromium |
| **Cu** | Copper |
| **Fe** | Iron |
| **K** | Potassium |
| **Mg** | Magnesium |
| **Mn** | Manganese |
| **Mo** | Molybdenum |
| **Na** | Sodium |
| **Ni** | Nickel |
| **P** | Phosphorus |
| **Pb** | lead |
| **S** | Sulfur |
| **Se** | Selenium |
| **Ti** | Titanium |
| **Zn** | Zinc |
| **DNA** | Deoxyribonucleic acid |
| **RNA** | Ribosomal ribonucleic acid |
| **OUT** | Operational taxonomic unit |
| **ANOVA** | Analysis of variance |
| **PERMANOVA** | Permutational multivariate analysis of variance |
| **PCoA** | Principle coordinate analysis |
| **LDA** | Linear discriminant analysis |
| **LEfSe** | Linear discriminant analysis effect size |

# ACKNOWLEDGEMENTS

The authors acknowledge Taylor Cass Adams and Sonia Tsai (USDA-ARS, Fayetteville, AR) Michelle Armstrong and Jenny Richter (USDA-ARS, Booneville, AR) and Lillian Meadors, Jamie Hess (University of Arkansas) for their help during the trial. Mention of trade names or commercial products in this manuscript is solely for providing specific information and does not imply recommendations or endorsement by the US Department of Agriculture. USDA is an equal opportunity provided and employer.

### Funding

The authors received no funding for this work.

### Competing Interests

The authors declare there are no competing interests.

### Author Contributions

- Mohan Acharya and Yichao Yang conceived and designed the experiments, performed the experiments, analyzed the data, prepared figures and/or tables, authored or reviewed drafts of the paper, and approved the final draft.
- Amanda J. Ashworth conceived and designed the experiments, analyzed the data, prepared figures and/or tables, authored or reviewed drafts of the paper, and approved the final draft.
- Joan M. Burke conceived and designed the experiments, performed the experiments, prepared figures and/or tables, authored or reviewed drafts of the paper, contributed on preparing research plots for the study, and approved the final draft.
- Jung Ae Lee analyzed the data, prepared figures and/or tables, authored or reviewed drafts of the paper, and approved the final draft.
- Roshani Sharma Acharya performed the experiments, prepared figures and/or tables, authored or reviewed drafts of the paper, contributed on preparing research plots for the study, and approved the final draft.

### DNA Deposition

The following information was supplied regarding the deposition of DNA sequences:
  Data are available at NCBI: PRJNA665918.

### Data Availability

  Raw data are available in the Supplemental Files.

### Supplemental Information

Supplemental information for this article can be found online at http://dx.doi.org/10.7717/peerj.11184#supplemental-information.

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
