# Peer review of "Soil microbial diversity in organic and non-organic pasture systems"

_PeerJ, doi:10.7717/peerj.11184_

## Round 0.1 · original submission · Major Revisions

Kindly address the comments made by the valuable reviewers.

Reviewer 1 ·

Basic reporting

No comments

Experimental design

No comments

Validity of the findings

No comments

Additional comments

This research focused on effects of organic pasture management on the soil microbiome for sustainable forage production as soil microbiome diversity contributes to improved nutrient cycling, soil structure, plant growth, and environmental resiliency. As previously soil microbes and their function under pasture grazing system was poorly investigated. I recommend this study for possible publication. Experimental planning using Illumina 16S rRNA gene amplicons followed by bioinformatics are enough to collect required data.

·

Basic reporting

The article entitled “soil microbial diversity in organic and non-organic pasture systems” present the response dynamic of soil microbial diversity to organic and non-organic livestock pastures using the 16S rRNA sequencing technique. Excellent work! Please see the attached word document; I have made some corrections within the text.

Experimental design

The authors described methods with sufficient detail and information to replicate; good work. However, the authors are advised to add weather data of the experimental site because changes in rainfall and temperature significantly influence the decomposition process in soil, which affects the soil microbial diversity under different field conditions.

Validity of the findings

All underlying data have been provided; they are robust, statistically sound, & controlled.

Additional comments

Specific Comments
Abstract
The abstract needs to be revised (e.g., give some specific results of the measured indices and name the methods to achieve the goal/aim described in the background).

Materials and Methods
The authors are advised to add weather data of the experimental site because changes in rainfall and temperature significantly influence the decomposition process in soil, which affects the soil microbial diversity under different field conditions.

Introduction
The introduction is not providing the enough information on the research background and research gap. The authors should add more literature on previous findings of soil microbial diversity in various conditions, especially under organic and non-organic pasture systems.

Discussion
I think authors should rethink what they write in the first paragraph, and only summarize the main findings in view of the research questions. After this, authors can explore in subsequent paragraphs different aspects of the work and explain how their findings expand the envelope of knowledge, but first of all, authors simply need to state the main findings without discussing their why and how or the relationships to the literature. First of all, the reader needs a clear statement on what the study found.

·

Basic reporting

no comment

Experimental design

no comment

Validity of the findings

no comment

Additional comments

The authors can add weather data if the authors want to add.

Reviewer 4 ·

Basic reporting

Acharya et al compared soil bacterial diversity between organic and inorganic pasture ecosystems and overall, paper is well written with relevant literature cited and I recommend minor revisions. I recommend authors to add more literature in introduction to strengthen the paper. Specifically, adding literature on effect of soil bacterial communities on management practices help since this paper aims at looking on the effect of soil bacterial community changes on management practices. Since grazing is the important component in experimental design, I also recommend authors to add relevant literature on grazing practices on grasslands and their effect on soil bacterial community composition and diversity. As established in the earlier sections about grazing, there were no relevant sections in the discussion about the effect of grazing on soil bacterial communities. There was literature supporting the effect of grazing and I highly recommend authors to revise and add effect of grazing on soil bacterial communities. I also recommend authors to discuss about the effect of plant species on soil bacterial diversity. In addition to the above mentioned, I have detailed other revisions in the following sections.

Experimental design

1. Do you have any baseline data on bacterial diversity and composition 10 years prior to the study?
2. Do you have data on bacterial or chemical composition shifts during this 10 years of time?
3. What is the rationale of sampling 2 years?
4. Aim implies study aimed at comparing changes in soil bacterial diversity over 10 years of time. However, the subsequent sections imply study of soil bacterial diversity on land that was organically managed for 10 years. I would suggest re-writing the sentence to be more clear for the readers.

Validity of the findings

1. Would the effect of bacterial diversity you have seen in organically managed grasslands would remain same if these are converted to croplands in future (organic management)?
2. Were the bacterial sequences of OTUs deposited in NCBI for access?
3. Proteobacteria are the dominant bacterial groups in general in most of the soil samples. Did you look at sub-classes like a-proteobacteria, b-proteobacteria and g-proteobacteria? Looking at sub-classes might give more insights. Please refer to Davinic, Marko, et al. "Pyrosequencing and mid-infrared spectroscopy reveal distinct aggregate stratification of soil bacterial communities and organic matter composition." Soil Biology and Biochemistry 46 (2012): 63-72.
4. In the lines 122-123 soil description was cited. Soil type greatly affect microbial composition. Did you see any relationship between soil type and bacterial diversity in literature with similar soils?
5. In the lines 202-203 it was said that greengenes database was used to classify OTU’s. Which bioinformatics pipeline was used to blast the sequences to classify?
6. In Figure 3 based on community composition differences, it is shown that samples collected in July and August are distinct in organic pasture lands. However, no such results were given in results section and also not discussed. Can you explain why soil bacterial community composition differed between july and august? Did you look if this effect is consistently seen in both the years or if any one particular year driving these changes? The bacterial diversity indices (Fig.2) were also distinct between july and august samples (organic pasture lands). Is there any particular bacterial group or sampling year driving these changes? Based on figure1, Bacteroidetes group were enriched in July (organic) samples compared to August. Bacteroidetes tend to grow under high resource rich enrichment. Please look into this and add supporting literature.

Additional comments

I recommend this paper for publication with minor changes as listed in the above sections.
1. How would this study help farmer’s community in future? How does your study help in taking management policies or decisions?
2. QIIME is the most commonly used pipeline for metagenomics data analysis. Caporaso, J. Gregory, et al. "QIIME allows analysis of high-throughput community sequencing data." Nature methods 7.5 (2010): 335-336.
My understanding is in this paper authors Acharya et al used several different bioinformatics tools to process and analyze sequencing data. Any specific reason for this approach instead of using most commonly used QIIME?
3. Lamb and sheep are interchangeably used in introduction and discussion sections. I would recommend consistency in using the terms.
4. In a recently published article it was shown that, rare microbiome has the potential of effecting soil functionality especially in fertilized lands. Did you look into the rare microbiome in your data set? Chen, Qing-Lin, et al. "Rare microbial taxa as the major drivers of ecosystem multifunctionality in long-term fertilized soils." Soil Biology and Biochemistry 141 (2020): 107686.
5. Suggest to rewrite line 391 (It is because their gram-positive cell wall 391 structure allows these organisms to withstand drier conditions).

---

## Round 0.2 · accepted · Accept

The authors have addressed all reviewers' comments and now the article should be considered for publication.

Reviewer 1 ·

Basic reporting

Authors compared soil bacterial diversity between organic and inorganic pasture ecosystems in this research. I already accept this paper. Paper is well written and now in revised version, paper is strengthened with more and latest literature on soil bacterial communities, their effect of plant species etc. I am satisfied and accept this paper to be published in Peer J.

Experimental design

Experimental planning using Illumina 16S rRNA gene amplicons followed by bioinformatics are enough to collect required data

Validity of the findings

This research focused on effects of organic pasture management on the soil microbiome for sustainable forage production as soil microbiome diversity contributes to improved nutrient cycling, soil structure, plant growth, and environmental resiliency. As previously soil microbes and their function under pasture grazing system was poorly investigated. I recommend this study for possible publication.

Additional comments

I recommend acceptance for possible publication in journal PeerJ.

·

Basic reporting

Literature references, sufficient field background/context provided.

Experimental design

Research question well defined, relevant & meaningful.

Validity of the findings

All underlying data have been provided; they are robust, statistically sound, & controlled.

Additional comments

Congratulations to all authors. I have read the revised version, and I am satisfied with the revision and response. Therefore, I recommend this paper for publication.

Regards,
Muhammad Ali Raza (Ph.D.)

·

Basic reporting

No comment

Experimental design

well explained and well structured.

Validity of the findings

conclusions are well linked to the study results and will be a good addition to the subject

Reviewer 4 ·

Basic reporting

Acharya et al compared soil bacterial diversity between organic and inorganic pasture ecosystems and overall, paper is well written with relevant literature cited and I recommend this research for publication. Authors have addressed all the questions of my review and made changes as per suggestions.

Experimental design

No Comment

Validity of the findings

No Comment